# Anti-chikungunya virus seroprevalence in Indigenous groups in the São Francisco Valley, Brazil

Jandir Mendonça Nicacio [1,2¤a]*, Ricardo Khouri [3,4,5¤b], Antônio Marconi Leandro da Silva[1¤a], Manoel Barral-Netto [3,4,6¤b], João Augusto Costa Lima [7], Ana Marice Teixeira Ladeia [8], Rodrigo Feliciano do Carmo [2,9], Anderson da Costa Armstrong [1,2¤a]*

**1** Federal University of Vale do São Francisco School of Medicine-UNIVASF; Petrolina, Pernambuco, Brazil, **2** Postgraduate Program in Health and Biological Sciences, Federal University of Vale do São Francisco-UNIVASF, Petrolina, Pernambuco, Brazil, **3** Oswaldo Cruz Foundation/Fiocruz, Institute Gonçalo Moniz, Salvador, Brazil, **4** Federal University of Bahia School of Medicine–UFBA; Salvador, Bahia, Brazil, **5** Rega Institute for Medical Research, KU Leuven, Leuven, Belgium, **6** Instituto Nacional de Ciência e Tecnologia de Investigação em Imunologia, São Paulo, Brazil, **7** Cardiology from Johns Hopkins University, Baltimore, Maryland, United States of America, **8** Postgraduate Course in Medicine and Human Health, Bahiana School of Medicine and Public Health, Salvador, Brazil, **9** College of Pharmaceutical Sciences, Federal University of Vale do São Francisco- UNIVASF, Petrolina, Pernambuco, Brazil

¤a Current address: Department of Medicine, Federal University of Vale do São Francisco- UNIVASF, Petrolina, Pernambuco, Brazil
¤b Current address: Oswaldo Cruz Foundation/Fiocruz, Institute Gonçalo Moniz, Salvador, Brazil
* jandir.nicacio@univasf.edu.br (JMN); anderson.armstrong@univasf.edu.br (ACA)

**Data Availability Statement:** All data, including identification of the indigenous people, cannot be shared publicly due to the information protection policy of the National Indian Foundation (FUNAI)

## Abstract

### Background

Chikungunya fever (CHIKF) is a serious public health problem with a high rate of infection and chronic disabling manifestations that has affected more than 2 million people worldwide since 2005. In spite of this, epidemiological data on vulnerable groups such as Indigenous people are scarce, making it difficult to implement public policies in order to prevent this disease and assist these populations.

### Objective

To describe the serological and epidemiological profile of chikungunya virus (CHIKV) in two Indigenous populations in Northeast Brazil, as well as in an urbanized control community, and to explore associations between CHIKV and anthropometric variables in these populations.

### Methodology/Principal findings

This is a cross-sectional ancillary study of the Project of Atherosclerosis among Indigenous Populations (PAI) that included people 30 to 70 years old, recruited from two Indigenous tribes (the less urbanized Fulni-ô and the more urbanized Truká people) and an urbanized non-Indigenous control group from the same area. Subjects underwent clinical evaluation and were tested for anti-CHIKV IgG by enzyme-linked immunosorbent assay. Serological

and the Special Indigenous Health Secretariat (SESAI) of Brazil. Regarding the data availability statement, as per guidance, we are providing contact details of a third, non-author third party who can make the data available, whenever necessary, while maintaining the anonymity of the indigenous people: Carlos Dorneles Freire de Souza, MSc, PhD Institutional e-mail: carlos. freire@arapiraca.ufal.br Affiliation: Federal University of Alagoas School of Medicine - UFAL, Arapiraca, Alagoas, Brazil The data can be found on the Special Indigenous Health Secretariat - SESAI and National Indian Foundation (FUNAI), Brazil. Link: https://saudeindigena.saude.gov.br/ and https://www.gov.br/funai/pt-br Part of the information, however, is published in the Supporting Information (supplementary material).

**Funding:** Conselho Nacional de Desenvolvimento Científico e Tecnológico-CNPq"- Ministry of Science, Technology, Innovations and Communications of Brazil (link: < https://www.gov.br/cnpq/pt-br>) provided logistical support and funding for the laboratory tests performed in the study. Funding was allocated to ACA. We acknowledge Fundação Maria Emilia for the grant regarding the publication fee and further research on this topic. The funders had no role in study design, data collection and analysis, decision to publish, or preparation of the manuscript.

**Competing interests:** The authors have declared that no competing interests exist.

profile was described according to ethnicity, sex, and age. The study population included 433 individuals distributed as follows: 109 (25·2%) Truká, 272 (62·8%) Fulni-ô, and 52 (12%) from the non-Indigenous urbanized control group. Overall prevalence of CHIKV IgG in the study sample was 49.9% (216; 95% CI: 45·1–54·7). When the sample was stratified, positive CHIKV IgG was distributed as follows: no individuals in the Truká group, 78·3% (213/272; 95% CI: 72·9–83·1) in the Fulni-ô group, and 5.8% (3/52; 95% CI: 1.21–16) in the control group.

## Conclusions/Significance

Positive tests for CHIKV showed a very high prevalence in a traditional Indigenous population, in contrast to the absence of anti-CHIKV serology in the Truká people, who are more urbanized with respect to physical landscape, socio-cultural, and historical aspects, as well as a low prevalence in the non-Indigenous control group, although all groups are located in the same area.

## Author summary

Chikungunya fever is a serious public health problem, with a high rate of infection and disease. Chikungunya virus (CHIKV) is a cosmopolitan virus, which has inflicted severe damage in 50 countries in the Americas and is responsible for chronic disabling manifestations. In spite of this, epidemiological data on vulnerable groups such as Indigenous people are scarce. We report on a cross-sectional study describing the seroprevalence of CHIKV in Indigenous groups in the São Francisco Valley, Brazil, in association with anthropometric data. The study population included 433 individuals distributed in the following ethnic groups: 109 (25.2%) Truká, 272 (62·8%) Fulni-ô, and 52 (12%) from the non-Indigenous urbanized control group When the sample was stratified, positive CHIKV IgG was distributed as follows: no individuals in the Truká group, 213/272 (78.3%; 95% CI: 72·9–83·1) individuals in the Fulni-ô group, and 3/52 (5.8%; 95% CI: 1·21–16) individuals in the control group. This study shows, for the first time, that CHIKV circulated in an Indigenous population (Fulni-ô) in the São Francisco Valley, in 2016 and 2017. The finding strikingly differs from the absence of anti-CHIKV serology found in the Truká people and from the low prevalence in the urban region of Juazeiro, Bahia.

## Introduction

The last decades have been marked by arbovirosis outbreaks with impact on public health. Among these, chikungunya fever (CHIKF) stands out, in the early 2000s with rapid spread, reaching global proportions [1]. Despite this, little is known about the behavior of this arbovirosis in restricted access groups, considering the ethnic characteristics of each population.

CHIKF, which is caused by the chikungunya virus (CHIKV), was silent for three decades until major outbreaks affected people in Kenya and India in 2004 and 2005, respectively [2,3]. In 2013, the disease reached the Americas, and it affected approximately 50 countries by the end of 2014, with almost 23,000 confirmed autochthonous cases [4,5]. In 2014, the disease officially arrived in Brazil, and the first autochthonous cases were registered, almost

simultaneously in the semi-arid region of the Brazilian Northeast (the poorest region in the country) and in the Amazon Forest Region [6]. Within about two years, nearly one million cases of CHIKF had been registered in the Americas, *Aedes aegypti* being the main vector [7,8]. However, the impact of CHIKV on Brazilian Indigenous populations is still unknown.

In Brazil, there are officially 760,350 Indigenous people, composing 416 ethnic groups and 6,238 villages, who receive health care assistance from 1,199 Indigenous Primary Health Units [9]. Indigenous groups are distributed throughout the vast territory of Brazil, living in various stages of urbanization. The Northeast Region of Brazil, where European colonization began, is home to Indigenous peoples with diverse ethnicities and traditions [10]. However, very few studies have evaluated the impact of new epidemics on Indigenous communities [11].

To date, in Brazil, the largest records of probable case reports of CHIKF were published in the years 2016 and 2017, with 277,882 and 185,605 cases, respectively, and more than 151,000 confirmed cases during these years. The states in the Northeast Region had the highest incidence rates in 2016 and 2017 (420·3 cases/100,000 inhabitants and 249·5 cases/100,000 inhabitants, respectively) [12]. In 2018, the Northeast Region showed a significant drop in notifications, becoming fourth out of five regions in terms of incidence (19·5 cases/100,000 inhabitants) [13]. In 2020, up to epidemiological week (EW) 46, the Northeast registered an incidence of 37·5 cases/100,000 inhabitants, showing an increase in cases compared to the previous year [14].

Currently, the Northeast state of Pernambuco, which registered new CHIKV case rates of 531·4 cases/100,000 inhabitants in 2016, has an incidence of 34 cases/100,000 inhabitants by EW 38 of 2020 [12,14]. These notifications, however, did not take the country's vulnerable Indigenous populations into consideration.

Seroprevalence studies on arbovirosis in Indigenous groups in the Americas are rare. In 2004, one study identified the seroprevalence of dengue IgG in the Yukpa and Barí Indigenous tribes in Venezuela [15]. Later, in 2015, a seroprevalence study of CHIKV, hantavirus, and rickettsia was conducted in the Tuchín Indigenous community in Northern Colombia, finding no cases of CHIKV in the study population [16].

This study describes the serological and epidemiological profile of CHIKV in two Indigenous populations in the Northeast Region of Brazil, at different stages of urbanization, in addition to an urbanized control community in the same region. Our hypothesis was that the groups with lower degrees of urbanization, socio-cultural confinement behavior, and conditions strongly related to worse sanitation and health education would have the highest prevalence of CHIKV. Furthermore, this study explores the associations between the serological profile of CHIKV and anthropometric variables usually related to worse outcomes in arbovirosis outbreaks [17].

## Methods

### Ethics statement

The research was approved by the National Research Ethics Council (CONEP number 1.488.268), the National Indigenous Foundation (Fundação Nacional do Índio [FUNAI]; process number 08620.028965/2015-66), and the Indigenous leaders of the participating groups. All participants provided written informed consent before taking part in the study [18,19].

### Study design and sample selection

This is a descriptive, cross-sectional, seroprevalence study, with stratified sample according to the degree of urbanization, carried out in specific populations of the São Francisco Valley in Northeast Brazil, composed of Indigenous groups from two tribes (Fulni-ô and Truká) and a non-Indigenous urbanized control group (from Juazeiro city, state of Bahia). The current

analysis was carried out as an ancillary study of the Project of Atherosclerosis among Indigenous Populations (PAI) study.

The PAI study protocol has been previously described in detail [18]. In summary, PAI was initiated in 2016 as an observational study, whose primary aim was to access cardiovascular health in two Indigenous groups with different degrees of urbanization and a totally urbanized control group. The PAI study included individuals of both sexes, 30 to 70 years old, without known cardiovascular disease or renal failure requiring hemodialysis, who provided authorization to conduct the study and signed the informed consent term [10,18]. The initial phase of the PAI study was completed in 2017, after including a total of 1061 participants: 321 from Fulni-ô tribe, 351 from Truká tribe, and 389 from the city of Juazeiro.

## Indigenous groups

The following three groups were included: one Indigenous tribe characterized by low level of urbanization (Fulni-ô, located on the banks of the Ipanema River in the São Francisco Basin); one tribe already affected by the urbanization process (Truká, whose territory is crossed by the northern axis of the São Francisco River Transposition Project); and a control group located in a city with a totally urbanized area and a low profile of migration in the municipality of Juazeiro, Bahia, which is also on the banks of the São Francisco River [18].

The Fulni-ô Indigenous tribe is located in the municipalities of Águas Belas and Itaíba, in the *agreste* region of the state of Pernambuco, and it comprises an area of 12,000 m$^2$ with 4,689 people, showing signs of less urbanization [18]. Once a year, the Fulni-ô people are required to be confined in isolation from non-Indigenous people for three months in a separate area adjacent to the tribe for the ritual known as *Ouricuri*. The Fulni-ô is the only Indigenous group in Northeast Brazil to maintain their original language for daily use [9,18].

The Truká group, on the other hand, comprises people located in the semi-arid region of the same state, in the middle of the São Francisco River; the population comprises 2,981 people, in an area of 6,000 m$^2$, in the municipality of Cabrobó, on the island known as Ilha de Assunção or Ilha Grande, which is part of the archipelago of Asunción, showing anthropological signs of greater urbanization [9,18]. They are known as a rural community. The geographical locations of the Truká, Fulni-ô, and control group are shown in Fig 1.

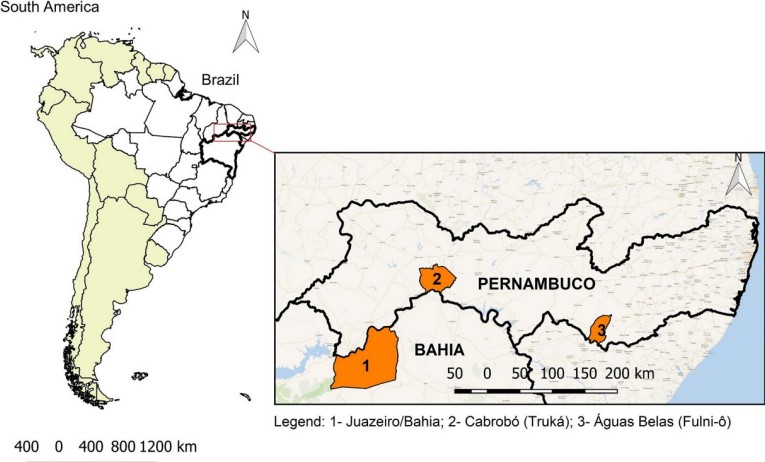

**Fig 1. Geographic locations of the Truká, Fulni-ô, and control group.** Map base layers were obtained from <http://www.naturalearthdata.com/about/terms-of-use/> covered by a Creative Commons Attribution 4.0 International (CC BY) License (https://creativecommons.org/licenses/by/4.0/legalcode). Map base layers were modified in QGIS software version 2.18.

The definition of urbanization in this study refers not only to population density, soil permeability to rainfall, scarcity of vegetation, and the characteristics of buildings, as defined by the Brazilian Institute of Geography and Statistics, but also to the historical and cultural aspects of the Indigenous groups [20]. In this context, the Fulni-ô group has no paved roads, and their houses are more isolated. Furthermore, they practice an annual three-month period of isolation from non-Indigenous peoples, in a ritual known as *Ouricuri*, and they preserve their original language, Yathê, in schools, along with Portuguese [18,19]. From this point of view, the Fulni-ô group was considered to be less urbanized. In contrast with the Fulni-ô, the Truká group presents intermediate characteristics of urbanization, with lifestyle changes and major infrastructural landscape transformation [18,19].

Both groups are assisted by a family group of health professionals, comprising a physician, a nurse, and a group of assistants and local community workers.

## Anthropometric data collection

Weight, age, height, and body mass index (BMI) were collected from August 2016 (EW 34) to June 2017 (EW 22), in two collection periods. During EW 34/2016, anthropometric data and biological samples were collected only in the Fulni-ô group. In EW 22/2017, a second moment of anthropometric data and biological sample collection was performed, not only in the Fulni-ô group, but also in the Truká tribe and in the control group. Data collection followed the criteria of the PAI Study [19]. For analysis, age was stratified as 30 to 40 years, 41 to 50 years, 51 to 60 years, and 61 to 70 years.

Obesity has been described as a major risk factor for severe clinical presentations of arbovirosis [17]. Therefore, we included obesity data in our analysis, as measured by BMI, according to the World Health Organization recommendations. In relation to BMI ($kg/m^2$), individuals were classified in a simplified manner as underweight ($< 18\cdot5$), normal ($\geq 18\cdot5$ and $< 25$), overweight ($\geq 25$ and $< 30$), and obese ($\geq 30$), in accordance with the World Health Organization [21].

## Collection of biological material

Peripheral blood collections were also performed between EW 34/2016 and EW 22/2017 with right or left antecubital fossa venipuncture. After appropriate antisepsis, 5 to 10 ml of venous blood were removed and transported in refrigerated boxes at 2 to 8˚C. Afterwards, serum was centrifuged in 1.5 ml Eppendorf tubes and conserved at −20 to −70˚C. These samples were stored at the Instituto Gonçalo Moniz, FIOCRUZ Bahia and the Laboratory of Clinical Analysis, LPC, Salvador, Bahia.

## Serological tests

Serological tests for anti-chikungunya virus IgG by enzyme-linked immunosorbent assay (ELISA) from Euromimmun (Code: EI 293a-9601G) were performed following manufacturer instructions. Values with relative index $\geq 1\cdot1$ were considered positive; values $\geq 0\cdot8$ and $< 1\cdot1$ were considered borderline, and values $< 0\cdot8$ were considered negative. Serological tests were performed on 451 participants of the PAI study who agreed to be included and who had viable serum samples at the time the test became available for the study, during 2019. Only 3 borderline samples from the Fulni-ô group were discarded. A total of 15 lipemic, icteric, hemolyzed, or failed samples were discarded from the 3 groups. No other serological tests were performed for other arboviruses in this study.

Although cross-reaction is a challenge in seroprevalence studies for arbovirus, it is particularly problematic in arboviruses of the same family, such as dengue fever, yellow fever, and,

more recently, Zika fever [22]. Regarding CHIKV, commercial ELISA diagnostic tests were evaluated, and they showed specificity and sensitivity of 95% and 88%, respectively, for IgG, with 18% false positive for IgM ELISA [23]. In addition, the EuroImmun anti-CHIKV ELISA IgG assay (EUROIMMUN AG, Lüebeck, Germany) showed 95·4% sensitivity and 100% specificity in validation studies on another specific population, in previous studies [24].

## Statistical analysis

Sample size calculation for the PAI study was previously described [18,19], computing a total of 957 participants (319 from each of the three groups). For this ancillary study, the calculated sample size was a total of 364 participants, considering a population of 7580 Indigenous people, 95% confidence interval (CI), 5% precision, and an estimated prevalence of CHIKV of 46%, based on a previous prevalence study for a municipality in the state of Bahia in December 2015 [25,26], using an online tool <http://sampsize.sourceforge.net/iface/>. Our hypothesis was that the groups with lower degrees of urbanization, socio-cultural confinement behavior and conditions strongly related to worse sanitation and health education would have the highest prevalence of CHIKV.

The choice of a municipality in Bahia (also located in the Northeast Region) as a reference for the prevalence estimated in the sample calculation was due to the absence of seroprevalence data in the regions close to the tribes. Furthermore, this municipality has climatic and demographic similarities with the cities where the Truká and Fulni-ô groups are located [27].

Data were entered into the SPSS computer program (SPSS Inc., Chicago, IL, USA, Release 16·0·2, 2008) twice, with automatic consistency and amplitude checking. Categorical variables were presented as absolute and relative frequencies, and continuous variables were presented as median (first–third quartiles) after verification of data normality by the Kolmogorov-Smirnov test. The frequency of infection by CHIKV was described in terms of percentage and respective 95% CI. In univariate analysis, positive and negative individuals were compared by Mann-Whitney U test and Pearson's chi-square ($\chi^2$) for continuous and categorical variables, respectively. The prevalence ratio was programmed to be calculated, using a multivariable model, only for those variables that showed a p value < 0·2 in the univariate analysis.

## Results

A total of 433 individuals with full available serological and clinical data were included; the majority were young women. Of the total, 124 (36·7%) individuals were considered obese, and 131 (38·8%) were considered overweight (Table 1).

The prevalence of anti-CHIKV IgG was 49·9% (216/433; 95% CI: 45·1–54·7). When evaluated separately by group, 3 (5·8%; 95% CI: 1·21–16), 213 (78·3%; 95% CI: 72·9–83·1), and 0 (zero) positive individuals were observed in the control, Fulni-ô, and Truká groups, respectively. When we evaluated the Fulni-ô individuals by collection period (EW 34/2016 and EW 22/2017), we observed that the percentage of individuals who were positive for anti-CHIKV IgG was maintained (79·6% and 78·1%, respectively).

Considering all participants, no significant associations were observed in CHIKV IgG seropositivity (n = 216; 49·9%) according to sex (male: 45·9%; female: 52·2%; p = 0·208) or age group (30 to 40 years: 52·3%; 41 to 50 years: 46·6%; 51 to 60 years: 48·7%; 61 to 70 years: 52·8%; p = 0·618). On the other hand, in relation to groups separated by degrees of urbanization, the Fulni-ô tribe showed a higher proportion of anti-CHIKV IgG in comparison to the Truká tribe and the control group, and this was not associated with sex, age group, or BMI categories (Table 2).

**Table 1. Description of characteristics of the total sample studied (n = 433).**

| Variables | N | % |
|---|---|---|
| *Sex* | | |
| Male | 159 | 36·7 |
| Female | 274 | 63·2 |
| *Group* | | |
| Control | 52 | 12·0 |
| Fulni-ô | 272 | 62·8 |
| Truká | 109 | 25·2 |
| **Age group** | | |
| 30–40 years old | 147 | 33·9 |
| 41–50 years old | 118 | 27·3 |
| 51–60 years old | 115 | 26·6 |
| 61–70 years old | 53 | 12·2 |
| **Nutritional status**[*] | | |
| Underweight | 1 | 0·3 |
| Normal | 82 | 24·1 |
| Overweight | 131 | 38·8 |
| Obese | 124 | 36·7 |
| **Variables** | **Median (1Q – 3Q)** | **Mean ± SD** |
| Age, years | 46·0 (38·0–55·0) | 46·8 ± 10·7 |
| Weight, kg | 72·6 (63·1–83·3) | 74·7 ± 16·5 |
| Height, cm | 160·0 (155·0–166·0) | 160·4 ± 8·1 |
| Body mass index, kg/m$^2$ | 28·1 (25·0–31·6) | 29·0 ± 5·9 |

SD- standard deviation; 1Q- first quartile; 3Q- third quartile.

[*]95 individuals missing weight and/or height data.

**Table 2. Prevalence of CHIKV IgG antibodies in population groups (control, Fulni-ô, and Truká) and association with age, sex, and anthropometric data.**

| Variables | CHIKV IgG | | |
|---|---|---|---|
| | **Control (n = 52)** | **Fulni-ô (n = 272)** | ***P*** |
| | **% (positive/total)** | **% (positive/total)** | |
| **Group prevalence** | 5·8 (3/52) | 78·3 (213/272) | <0·001 |
| **Sex** | | | |
| Male | 0·0 (0/26) | 75·2 (73/97) | 0·213 |
| Female | 11·5 (3/26) | 80 (140/175) | |
| **Age Group** | | | |
| 30–40 years | 5·9 (1/17) | 78·4 (76/97) | 0·650 |
| 41–50 years | 6·7 (1/15) | 81·8 (54/66) | |
| 51–60 years | 0·0 (0/11) | 75·7 (56/74) | |
| 61–70 years | 11·1 (1/9) | 77·1 (27/35) | |
| **BMI Group**[*] | | | |
| Underweight | 0·0 (0/0) | 100 (1/1) | 0·372 |
| Normal | 0·0 (0/12) | 66·7 (32/48) | |
| Overweight | 6·3 (1/16) | 45·8 (64/77) | |
| Obese | 9·1 (2/22) | 80·0 (44/55) | |

BMI- Body mass index

[*]72 CHIKV IgG+ individuals with missing weight and/or height, all from the Fulni-ô group. Pearson's chi-square ($\chi2$) was used to calculate p.

Considering participants anti-CHIKV IgG seropositives, there were no significant differences in median age between Fulni-ô people and urbanized controls (47·0; IQR: 37·0–54·0 years vs. 49·0; IQR: 41·0–59·0 years, respectively; p = 0·528), or regarding median values of BMI (28·1; IQR: 25·2–30·7 kg/m$^2$ vs. 30·1; IQR: 29·8–32·5 kg/m$^2$, respectively; p = 0·354). Also, there was no significant difference in median age or median values of BMI when assessing anti-CHIKV IgG seropositive and negative participants in these groups (S1 Table).

## Discussion

Our study shows a high seroprevalence of CHIKV in one traditional Indigenous ethnicity, out of three study groups with different stages of urbanization and diverse environmental conditions. Our results suggest that the most traditional, less urbanized Indigenous community was the one most exposed to CHIKV, as demonstrated by the highest serological rate of infection.

Variable results of anti-CHIKV seropositivity have been found in urbanized areas in Brazil. A study evaluated the prevalence in two cities in the state of Bahia, Feira de Santana and Riachão do Jacuípe, showing a total seropositivity of 51% in December 2015 [25], whereas, in Chapada, a district of Riachão do Jacuípe, a rate of 20% positivity was observed in April 2016, and 11·8% seropositivity was observed in a community of Salvador, Bahia between November 2016 and January 2017 [28,29]. An epidemiological study evaluating two nearby regions, with the same climatic conditions during the same EWs, found different prevalences of 57·1% and 45·7% [25]. This difference in seroprevalence in areas with similar climatic and geographical characteristics is possibly related to several factors, such as population density, sampling type, degree of vector mosquito infestation, sanitary structure of neighborhoods, and commercial activity in the city [10,11,25].

The present report draws attention to the fact that CHIKV seems to have had a particular impact on the Fulni-ô people, the least urbanized Indigenous group in the Northeast Region of Brazil, showing more than 70% seroprevalence, in contrast to the absence of anti-CHIKV IgG in the Truká tribe. This significant and unexpected result in the Fulni-ô tribe differs from most epidemiological studies in the Americas, where estimated and found prevalence has been less than 50% [28–30]. To date, few studies have recorded such high prevalences. Among those that have, we highlight 75% in Comoros Island in 2004, 63% in Kenya in 2005 [31,32], and 75·6% in Haiti, in children aged 2 to 14 years, in 2014 [33]. It is, however, important to mention that these studies were conducted in non-Indigenous communities, with different degrees of urbanization.

Interestingly, our findings suggest that the more traditional Indigenous tribes might have very particular epidemiological dynamics, regarding viral outbreaks. On the other hand, when assessing non-Indigenous areas around the Fulni-ô tribe (Águas Belas and Itaíba), official data indicate a low rate of notification of suspected cases based on clinical evaluation, considering the cumulative period of 2015 to 2020. Importantly, there are no official data on serological prevalence of CHIKV in this area. The municipality of Águas Belas has a population of 40,235 people, and it reported a total of 167 suspected cases in 2016, remaining below 12 cases/year in subsequent years (2017 to 2020). The municipality of Itaíba (total population of 26,256 people), which is adjacent to Águas Belas, reported 143 cases of CHIKV between the years 2015 and 2017. In 2016, of the 167 cases of suspected CHIKV notified in Águas Belas, 55% were confirmed by laboratory and/or clinical epidemiological criteria. Of the total of 11 cases notified in Itaíba for 2017, only 1 case was confirmed by serological testing. The raw data for the non-Indigenous municipalities were acquired from the official epidemiological vigilance and are available as Supplementary Material (S1 and S2 Appendices).

Regarding the degree of urbanization in isolated communities, our study also differs from previous assessment of seroprevalence of dengue in the Yukpa (most influenced by urbanization, concerning aspects of the physical landscape, house structure, sanitation conditions, as well as population traffic) and Barí Indigenous groups, in Sierra de Perija, Venezuela, which found higher prevalence in the most urbanized tribe, with greater transit of individuals with the virus within the community, precarious health structure, and higher infestation of *Ae. Aegypti* [15].

There are no official data for vector mosquito presence that are specific for the studied tribes. Nevertheless, since 2002, the Brazilian Ministry of Health has been conducting a survey in the nearby cities (*Ae. aegypti* Infestation Index Rapid Survey LIRAa). It consists of a method of monitoring the levels of infestation of *Ae. aegypti* larvae in homes, performed by periodical sampling technique, allowing an entomo-epidemiological survey of the region [34]. In 2017, Águas Belas showed an *Ae. aegypti* Infestation Index Rapid Survey of 2.0, which is considered alert, while, in the city of Cabrobó (adjacent to where the Truká tribe is located), the index was 0·2, which is considered low. In the same period, the city of Juazeiro (control group) showed an Index of 0·8 (considered low or satisfactory) [34]. The different levels of infestation of the vector mosquito might partially explain the higher prevalence of CHIKV in the Fulni-ô people, when compared to the Truká people.

Another important aspect for our finding of diverse CHIKV prevalence ratios may be the specifics of weather conditions in different regions. While the tribes are located along the São Francisco River Valley, in an area with tropical climate, the Fulni-ô group is located in the *agreste* region, where rainfall is slightly higher than in the Truká group area; this may favor greater vector mosquito infestation. [35]. However, this does not fully explain our results, given that other cities with similar weather conditions show variable notifications of suspected CHIKF cases in the same period [36].

There has been much discussion about the role of poorly planned urbanization and disorderly growth in the health of the population, especially with regard to contagious or insect borne diseases [37]. The high prevalence of CHIKV in the Fulni-ô people may be favored by the degree of urbanization, with unpaved streets, houses that preserve original constructions, which are partly made of clay, and less sanitary structure [9,18]. The Fulni-ô people also struggle for access to adequate water supply, and they usually store still water, what might be related to mosquito proliferation. This association of CHIKV with the condition of socioeconomic vulnerability has already been identified in previous studies in the Americas, including in a non-Indigenous community in the Brazilian Northeast [28,38]. These factors, in conjunction with a lifestyle characterized by social coexistence, with strong cultural exchange between families and the preservation of confinement rituals (*Ouricuri*), have probably influenced the high rate of infectivity and transmissibility, since people live close together with multiple close social interactions, in the same environment of the mosquito vector, making the Fulni-ô Indigenous Reserve an environment conducive to maintaining humans in the viral cycle over the years [9,18].

This study has important limitations that must be considered. Serology was not conducted for other arboviruses that are prevalent in Brazil, and potential serological cross-reactions could not be addressed. The sample was not homogeneous with respect to the proportion of participants in each subgroup. There was a gap between the blood sample collection and the serum analysis. We assessed all blood samples that were properly stored in a sufficient amount for serological testing. The number of tested participants exceeded the sample size estimate, but a relevant number of participants from the main PAI study were not included in this ancillary analysis. Furthermore, there was a lack of clinical data to analyze the association of signs and symptoms with seroprevalence data; thus, we could not address the entire dimension of

the clinical relevance related to the chikungunya disease burden in these Indigenous populations. It is, however, necessary to highlight that working with closed populations, who are distant from large urban centers and who have distinct social behaviors, makes it difficult to execute projects in this profile, in addition to the intrinsic logistical limitations to approach and data collection in more isolated populations of difficult access.

Chikungunya compromises quality of life, interfering with social and work relationships and often leading to psychiatric disorders [39,40]. This acutely debilitating febrile illness is a serious public health problem, with the risk of evolving into chronic forms. Indigenous people in the Americas have been through known recurrent infectious diseases outbreaks since colonial times, but little is known regarding the dynamics of the disease, as well as the long-term impact on these populations [41]. It is important to understand the epidemiological dynamics of chikungunya, as well as other new infectious diseases, in traditional vulnerable populations such as Indigenous people, in order to aid public health policies directed to the management of chronic forms of the disease and the control of future outbreaks.

In conclusion, our study shows a high serological prevalence for CHIKV in a traditional Indigenous population (Fulni-ô) in the São Francisco Valley between 2016 and 2017, regardless of age and sex. This finding differs from the low prevalence found in the more urbanized Truká people and in the non-Indigenous urban control group, although all communities are located in the same region. Understanding the dynamics of epidemics in various locations and their peculiarities in diverse population groups will allow for more appropriate planning of prevention and containment of future epidemics.

## Supporting information

**S1 Appendix. Notification of suspected cases of chikungunya fever in Águas Belas municipality, from 2016 to 2020.**
(XLSX)

**S2 Appendix. Notification of suspected cases of chikungunya fever in Itaíba municipality, from 2015 to 2017.**
(XLSX)

**S1 Table. Analysis of median values of age and BMI with anti-chikungunya virus seroprevalence.**
(DOCX)

## Acknowledgments

We would like to thank the Executive Secretariat of Health Surveillance, particularly the members of the team of the Arboviruses Surveillance Management of the State of Pernambuco and V Regional Health Management for their collaborative efforts on the interpretation of the official data for non-Indigenous areas. We also thank Acácio Willian Faustino de Andrade for helping us to edit the figure referring to the location map of the Fulni-ô and Truká tribes.

## Author Contributions

**Conceptualization:** Jandir Mendonça Nicacio, Antônio Marconi Leandro da Silva, João Augusto Costa Lima, Ana Marice Teixeira Ladeia, Anderson da Costa Armstrong.

**Data curation:** Ricardo Khouri, Antônio Marconi Leandro da Silva, Manoel Barral-Netto, Rodrigo Feliciano do Carmo, Anderson da Costa Armstrong.

**Formal analysis:** Rodrigo Feliciano do Carmo, Anderson da Costa Armstrong.

**Funding acquisition:** Anderson da Costa Armstrong.

**Investigation:** Jandir Mendonça Nicacio, Ricardo Khouri, Antônio Marconi Leandro da Silva, Manoel Barral-Netto, João Augusto Costa Lima, Ana Marice Teixeira Ladeia, Rodrigo Feliciano do Carmo, Anderson da Costa Armstrong.

**Methodology:** Jandir Mendonça Nicacio, Ricardo Khouri, Manoel Barral-Netto, Rodrigo Feliciano do Carmo, Anderson da Costa Armstrong.

**Resources:** Anderson da Costa Armstrong.

**Supervision:** Ricardo Khouri, Manoel Barral-Netto, Rodrigo Feliciano do Carmo.

**Writing – original draft:** Jandir Mendonça Nicacio, Ana Marice Teixeira Ladeia, Rodrigo Feliciano do Carmo, Anderson da Costa Armstrong.

**Writing – review & editing:** Jandir Mendonça Nicacio, Ricardo Khouri, Antônio Marconi Leandro da Silva, Manoel Barral-Netto, João Augusto Costa Lima, Rodrigo Feliciano do Carmo, Anderson da Costa Armstrong.

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
