## [Decision Letter · Decision Letter 0]

14 Dec 2020

Dear Dr Nicacio,

Thank you very much for submitting your manuscript "Anti- chikungunya virus seroprevalence in indigenous groups in the São Francisco Valley, Brazil" for consideration at PLOS Neglected Tropical Diseases. As with all papers reviewed by the journal, your manuscript was reviewed by members of the editorial board and by several independent reviewers. In light of the reviews (below this email), we would like to invite the resubmission of a significantly-revised version that takes into account the reviewers' comments. 

We cannot make any decision about publication until we have seen the revised manuscript and your response to the reviewers' comments. Your revised manuscript is also likely to be sent to reviewers for further evaluation.

Sincerely,

Brett M. Forshey

Associate Editor

Eugenia Corrales-Aguilar

Deputy Editor

The reviewers provided a number of issues to address to improve the manuscript, including:

- The Methods section needs to be clarified in a number of places, as mentioned below. For example, there should be more description of the PAI study and the sampling methodology.

- The interpretation of the data needs to be better justified in some places, including the mention that this population is most endangered for new viral outbreaks. In addition, the data just seems surprising and the reasons for the outcome should be explored a bit more - the prevalence is extremely high in one relatively remote community, but zero or nearly zero in communities you'd expect to have higher exposure. Also, how many acute cases were reported from these regions? >78% attack rate is quite high - was there a recognized outbreak in recent years? Hard to imagine a 78% attack rate would've gone unnoticed, but there's no context provided here.

Other issues to address, in addition to the comments by the reviewers:

- The concept of a 'control' population is not clear in the manuscript. In what way would a more urbanized population be a control? Where these anticipated to be more highly positive? The hypothesis being tested that would make this population the control is not clear and should be explained.

- Line 10: Mentions a million cases in Kenya and India, but the references don't actually speak to those numbers. Were there really a million cases in Kenya?

- Line 23: "CHKF"

- Lines 49-51, Ethical issues: The PAI study was approved by the various stakeholders, but was this add-on CHIKV study specifically approved?

- Lines 58-63: A map might be helpful in explaining how these communities are situated relative to one another

- Methods: what was the timing of the sampling? Line 73 says there were two collection periods, but it wasn't clear which groups were sampled when.

- Lines 89-90: How many samples were discarded? How did that impact overall participant numbers?

- Lines 108-110: The wording in these sentences is odd. I don't usually see 'prevalence' represented first as a count number. I recommend referring to the proportion first, with the total number of positives in parentheses. E.g. "The prevalence of anti-CHIKV IgG in the study sample was 49.9% (216/433; 95% CI: 45.1 - 54.7)"

- Lines 115-116: The comment that anti-CHIKV antibodies were more common in females than males (66.2% vs 33.8%) is very confusing - this is largely just a function of the fact that more females participated in the study. Along these same lines, on Table 2, it would be easier to interpret if the data were presented as the percent antibody positive rather than the percent of each category. That is, the number in the bracket should be the percentage of each group/variable that are positive or negative for anti-CHIKV antibodies.

- Lines 128-130 and Table 3: I don't follow how the prevalence ratio calculation was approached. What was it adjusted for? How is Truka the reference category when the prevalence is zero? How should these numbers be interpreted?

- Lines 136-138: This statement is a repeat from earlier in the Results.

- Lines 167-175: This paragraph needs a great deal of additional clarification - the paragraph either needs to be rewritten or deleted entirely. How is "climate" involved here? What was the timing of the mosquito survey? What is a rainfall index and how does it relate to Aedes abundance? I would expect that all of these locations could harbor Aedes mosquitoes, so this paragraph should focus on explaining what mosquito surveys were done at all 3 locations and when, with what regularity, and how that might relate to understanding CHIKV transmission. Aedes populations can be quite dynamic, so a single survey might not really capture transmission potential.

- Lines 193-195: Mentions the long interval between sample collection and testing - please clarify. How long? In what way would this interfere with the results?

- Lines 193-200, Limitations: Recommend mentioning potential for serological cross-reaction.

Reviewer's Responses to Questions

**Key Review Criteria Required for Acceptance?**

**Methods**

-Are the objectives of the study clearly articulated with a clear testable hypothesis stated?

-Is the study design appropriate to address the stated objectives?

-Is the population clearly described and appropriate for the hypothesis being tested?

-Is the sample size sufficient to ensure adequate power to address the hypothesis being tested?

-Were correct statistical analysis used to support conclusions?

-Are there concerns about ethical or regulatory requirements being met?

Reviewer #1: In line 48 from the Methods section, the authors should provide more explication about the choice of the 30 to 70 years of age range for recruitment. 

In line 48, the Authors should clearly state the approval number from Research Ethics Committee, regarding the work with Traditional Populations. 

The authors used a commercial kit for detection of anti-chikungunya virus IgG by enzyme-linked immunosorbent assay. Since it is widely known the occurrence of cross-reactions in serological tests to detect arbovirus infections, the authors should provide more details whether other tests were performed to detect other arbovirus species, or at least discussing the performance data from the kit used in the study. 

In line 90, the authors should clearly provide the exact number of samples discarded.

Reviewer #2: The methodology of this ancillary study is well described; however, it is necessary to state the sampling strategy. Also, I suggest clarifying the term “stratified” in line 44: does it refer to stratification in the context of a complex sampling design of the PAI study or to the three exposure groups? The exposure and control groups were adequately selected in terms not only of ethnicity but also of a gradient of urbanization, which allowed the evaluation of the contrasts of main interest; however, it is not clear the reason(s) behind testing a hypothesis of association between body mass index (BMI) and seroprevalence (lines 40-41): I suggest to provide evidence in support of the biological plausibility and/or clinical relevance of such relationship. The statistical approach seems to be consistent with the aim of the study, however, it is not clear whether the PAI study (and the ancillary) had a complex sampling design and, therefore, whether or not point and uncertainty estimates should account for it: Please provide further details.

**Results**

-Does the analysis presented match the analysis plan?

-Are the results clearly and completely presented?

-Are the figures (Tables, Images) of sufficient quality for clarity?

Reviewer #1: In table 1, there is a typing error in the "Mean DP", perhaps it might be "SD- standard deviation".

Reviewer #2: Results from descriptive statistics and univariate analysis are clear and correspond to the analysis plan, under de assumption that PAI (and the ancillary) study did not followed a complex sampling design (i.e., simple random sampling): Please, consider to change the title of table 1 (“Description of anthropometric parameters (n = 433)”) as it shows more than anthropometric parameters for which there is a high proportion of missing data (22.2%). I have the following concerns about the multivariate analysis: 1. What was the control group? If this group was the “urbanized control community” as its denomination suggests, then is incorrect to present a prevalence ratio for that group (Table 3: APR=0.966), furthermore, an APR lower than the unity (0.608) for the Fulni-ô is not plausible since prevalence in this group is 78.3% (213/272) as compared to 5.8% (3/52) in the “urbanized control community”; 2. Alternatively, if the “control” group was the Truká, then there would be a zero prevalence in the denominator of any PR (or APR); and 3. There is no indication of which variables the prevalence ratio was adjusted for or what were the reasons behind selecting them to be included in a multivariate model (the univariate analysis does not support a relationship to prevalence for any covariate).

**Conclusions**

-Are the conclusions supported by the data presented?

-Are the limitations of analysis clearly described?

-Do the authors discuss how these data can be helpful to advance our understanding of the topic under study?

-Is public health relevance addressed?

Reviewer #1: The authors claim that the "less urbanized Indigenous community might be the most endangered for new viral outbreaks". This sound a very strong statement, since the authors provide information about only one arbovirus species, and the prevalence of CHIKV infections in the indigenous populations analyzed indicated that this population likely presents herd immunity to CHIKV. The authors should reformulate the sentence.

Reviewer #2: In general, the validity of a conclusion regarding prevalence is contingent upon the sampling design, something that the authors did not describe in the manuscript and cannot be traced from the references to the main study (PAI). Furthermore, there are concerns about the analysis itself that must be clarified before attempting an interpretation of the results.

**Editorial and Data Presentation Modifications?**

Reviewer #1: (No Response)

Reviewer #2: N.A.

**Summary and General Comments**

Reviewer #1: The authors recruited patients from indigenous Populations in the São Francisco Valley, Brazil between August 2016 and June 2017, to collect clinical data and blood samples that underwent serology testing for anti-CHIKV IgG by ELISA. By analyzing the data from 433 patients, the authors argues that there is a high prevalence of Chikungunya virus infection in the Fulni-ô indigenous population between 2016 and 2017. The study found 78·3% of the samples from Fulni-ô group tested positive for anti-CHIKV IgG. Authors' conclusions are consistent with the results presented in the study, they provide a discussion about limitations of the study as well as on previously published related literature. 

In line 13 from the introduction section, the authors should provide a reference for the claim: "in the semi-arid region of the Brazilian Northeast (the poorest region in the country) and in the Amazon Forest region. "

Reviewer #2: Nicacio et al., conducted a Chikungunya serosurvey in two indigenous groups and in an urbanized control community (30-70 years old) in Northeast Brazil, as an ancillary study of the Project of Atherosclerosis among Indigenous Populations (PAI). This is a relevant research question because of the particularities and vulnerability of the target populations; however, the absence of a detailed description of the methodological approach – specifically, the sampling design – of both the principal and the ancillary studies precludes to reach meaningful inferences regarding the burden of the infection as determined by estimates of seroprevalence.

PLOS authors have the option to publish the peer review history of their article (what does this mean?). If published, this will include your full peer review and any attached files.

Reviewer #1: No

Reviewer #2: No
---

## [Decision Letter · Decision Letter 1]

9 Mar 2021

Dear Dr Nicacio,

Thank you very much for submitting your manuscript "Anti- chikungunya virus seroprevalence in indigenous groups in the São Francisco Valley, Brazil" for consideration at PLOS Neglected Tropical Diseases. As with all papers reviewed by the journal, your manuscript was reviewed by members of the editorial board and by several independent reviewers. In light of the reviews (below this email), we would like to invite the resubmission of a significantly-revised version that takes into account the reviewers' comments. 

We cannot make any decision about publication until we have seen the revised manuscript and your response to the reviewers' comments. Your revised manuscript is also likely to be sent to reviewers for further evaluation.

Sincerely,

Brett M. Forshey

Associate Editor

Eugenia Corrales-Aguilar

Deputy Editor

The authors made a number of improvements to the manuscript in response to the reviews. However, there are still significant issues to address, as indicated by the reviewer and by my comments below. I'm also attaching a track changes version of the manuscript with a number of other comments.

- Abstract: The confidence interval for the Fulni-o group does not include the point estimate.

- Abstract and elsewhere: It's not clear what makes one tribe more "urbanized" than another - that term is never defined, and the population density of the two tribes seems similar.

- Introduction: The first paragraph of the Introduction about pandemics is too generic and does not add to the manuscript -recommend deleting.

- Introduction line 10-11: States that South and Central America had 1.5 million chikungunya cases in 2013, but that does not seem to be correct.

- Figure 1: the 'control' area should be shown on the map as well.

- Methods, line 102: States that "Peripheral blood collections were also performed in EW 22/2017" which is confusing. Since the section is titled Collection of biological material, all collections should be mentioned here.

- Methods, lines 125-128: The sample size calculation needs to be explained better. What hypothesis was being tested - was this to compare populations? Estimate the overall prevalence? With what precision? Etc.

- Table 2: This table should show the breakdown of CHIKV seropositivity by variable (eg by age group), not just the overall percentage of participants by age group. So, what percentage of females were seropositive and what percentage of males were seropositive? Also please include confidence intervals.

- Discussion: in multiple places the authors refer to "official data indicate a markedly lower prevalence of CHIKV in the non-indigenous areas" - but no seroprevalence data is shown to compare. Please clarify and provide a reference to the data.

There are other comments and recommendations to consider in the word version of the submission, attached.

Reviewer's Responses to Questions

**Key Review Criteria Required for Acceptance?**

**Methods**

-Are the objectives of the study clearly articulated with a clear testable hypothesis stated?

-Is the study design appropriate to address the stated objectives?

-Is the population clearly described and appropriate for the hypothesis being tested?

-Is the sample size sufficient to ensure adequate power to address the hypothesis being tested?

-Were correct statistical analysis used to support conclusions?

-Are there concerns about ethical or regulatory requirements being met?

Reviewer #2: In this revised version of the manucript the authors clarified most of the issues raised before, for example, the reason behind the selection of the control group and the sampling of the ancillary study. Noticebly, the authors acknowledged the error made in estimating prevalence ratios (PR); however, it is not clear to me why didn't they re-estimate the PR (at least for the group with non zero prevalence). I suggest to provide additional information about participation rates by group and if possible, also to include a contrast between the samples of the ancillary (n=433) and main (n=1061) studies. This information might help the authors to account for the non expected results as declared in the discussion section.

**Results**

-Does the analysis presented match the analysis plan?

-Are the results clearly and completely presented?

-Are the figures (Tables, Images) of sufficient quality for clarity?

Reviewer #2: The results correctly reflect the analysis plan and are clearly presented.

**Conclusions**

-Are the conclusions supported by the data presented?

-Are the limitations of analysis clearly described?

-Do the authors discuss how these data can be helpful to advance our understanding of the topic under study?

-Is public health relevance addressed?

Reviewer #2: The conclusions are supported by the data, however, as I pointed out before (methods), providing further description of the sample that participated in the study might shed light to the interpretation of some unexpected findings, as declared by the authors, or at least to rule out the effect of selection bias due to diferential participation rates across groups.

**Editorial and Data Presentation Modifications?**

Reviewer #2: N.A.

**Summary and General Comments**

Reviewer #2: N.A.

PLOS authors have the option to publish the peer review history of their article (what does this mean?). If published, this will include your full peer review and any attached files.

Reviewer #2: No
---

## [Editor Report · Decision Letter 2]

14 Apr 2021

Dear MD Nicacio,

Thank you very much for submitting your manuscript "Anti- chikungunya virus seroprevalence in indigenous groups in the São Francisco Valley, Brazil" for consideration at PLOS Neglected Tropical Diseases. As with all papers reviewed by the journal, your manuscript was reviewed by members of the editorial board and by several independent reviewers. The reviewers appreciated the attention to an important topic. Based on the reviews, we are likely to accept this manuscript for publication, providing that you modify the manuscript according to the review recommendations. 

Sincerely,

Brett M. Forshey

Associate Editor

Eugenia Corrales-Aguilar

Deputy Editor

The authors have done a good job of addressing the reviewers concerns, with one major exception: Table 2 still does not show the data necessary for this publication. This should be the breakdown of participants and seropositivity, by subpopulation, by region. For example, it should show of the number of males from the control region, how many were positive and what percentage. Same for females from the control region, same for males from the indigenous regions, etc; the same breakdown should be shown for other variables such as age, etc. I've tried to demonstrate one way of displaying this data in track changes in the word document the authors had provided (attached). Also, here are other publications showing the data along the lines of what is needed for this manuscript - it doesn't have to be this format exactly, but hopefully will give you the idea of the data needed.

Table 1 in this manuscript: https://journals.plos.org/plosntds/article?id=10.1371/journal.pntd.0008355

Table 1 in this manuscript: https://journals.plos.org/plosntds/article?id=10.1371/journal.pntd.0006163

Please also clarify what the data at the bottom of Table 2, for age and BMI, represent - is that the mean age and BMI for positive individuals from those regions? How do those numbers compare with negative individuals from the same regions?

Figure Files:

Data Requirements:

Reproducibility:

References

---

## [Editor Report · Decision Letter 3]

11 May 2021

Dear MD Nicacio,

We are pleased to inform you that your manuscript 'Anti- chikungunya virus seroprevalence in Indigenous groups in the São Francisco Valley, Brazil' has been provisionally accepted for publication in PLOS Neglected Tropical Diseases.

Best regards,

Brett M. Forshey

Associate Editor

Eugenia Corrales-Aguilar

Deputy Editor

---

## [Editor Report · Acceptance letter]

14 Jun 2021

Dear MD Nicacio,

We are delighted to inform you that your manuscript, "Anti- chikungunya virus seroprevalence in Indigenous groups in the São Francisco Valley, Brazil," has been formally accepted for publication in PLOS Neglected Tropical Diseases.

Best regards,

Shaden Kamhawi

co-Editor-in-Chief

Paul Brindley

co-Editor-in-Chief
